# Influences of Software Changes on Oxycodone Prescribing at an Australian Tertiary Emergency Department: A Retrospective Review

**DOI:** 10.3390/pharmacy12020044

**Published:** 2024-03-01

**Authors:** Giles Barrington, Katherine Davis, Zach Aandahl, Brodie-Anne Hose, Mitchell Arthur, Viet Tran

**Affiliations:** 1Royal Hobart Hospital, Tasmanian Health Service, Hobart 7000, Australia; brodie.hose@ths.tas.gov.au (B.-A.H.); mitchell.arthur@health.tas.gov.au (M.A.); v.tran@utas.edu.au (V.T.); 2Austin Health, Heidelberg 3084, Australia; katherineadavis@icloud.com; 3School of Natural Sciences, University of Tasmania, Hobart 7000, Australia; zach.aandahl@utas.edu.au; 4School of Medicine, University of Tasmania, Hobart 7000, Australia; 5Menzies Institute for Medical Research, University of Tasmania, Hobart 7000, Australia

**Keywords:** prescribing, opioid, prescribing software, nudge theory, forcing function

## Abstract

Opioid prescribing and dispensing from emergency departments is a noteworthy issue given widespread opioid misuse and diversion in many countries, contributing both physical and economic harm to the population. High patient numbers and the stochastic nature of acute emergency presentations to emergency departments (EDs) introduce challenges for prescribers who are considering opioid stewardship principles. This study investigated the effect of changes to electronic prescribing software on prescriptions with an auto-populated quantity of oxycodone immediate release (IR) from an Australian tertiary emergency department following the implementation of national recommendations for reduced pack sizes. A retrospective review of oxycodone IR prescriptions over two six-month periods between 2019 and 2021 was undertaken, either side of a software adjustment to reduce the default quantities of tablets prescribed from 20 to 10. Patient demographic details were collected, and prescriber years of practice calculated for inclusion in linear mixed effects regression modelling. A reduction in the median number of tablets prescribed per prescription following the software changes (13.5 to 10.0, *p* < 0.001) with little change in the underlying characteristics of the patient or prescriber populations was observed, as well as an 11.65% reduction in the total number of tablets prescribed. The prescriber’s years of practice, patient age and patient sex were found to influence increased prescription sizes. Reduced quantity of oxycodone tablets prescribed was achieved by alteration of prescribing software prefill parameters, providing further evidence to support systems-based policy interventions to influence health care providers behaviour and to act as a forcing function for prescribers to consider opioid stewardship principles.

## 1. Introduction

Pharmaceutical opioids are prescribed for the treatment of moderate to severe acute and malignant pain [1]. These medications are a mainstay of emergency medical practice, playing an important role in providing pain relief to many people during an acute episode of care [2]. A multitude of opioids are available for administration through parenteral, transdermal and enteral routes. Of these, immediate release (IR) oxycodone, a semi-synthetic opioid used for the oral administration of strong pain relief, is the most frequently dispensed opioid in Australia [3]. While effective and widely used, there is an elevated risk profile for oxycodone IR as with other opioids, which includes inadvertent overdose, diversion, opioid dependence and addiction. The impact of increased availability of prescription opioids has been well-documented in the United States (US) where an increased pressure to overtreat pain and the aggressive marketing of opioids has led to a crisis of opioid use disorder, addiction and related deaths, with opioid poisoning now a leading cause of death for Americans under 50 [4]. The escalation in American opioid-related deaths is attributed to both an increase in prescription rates of pharmaceutical opioids from the mid-1990s, peaking in 2011 [5], as well as a surge in the use of illicit drugs such as heroin and fentanyl [4]. There is significant variance within the United States with regards to the prescription of pharmaceutical opioids, with differences observed within counties and states and in the differential availability of nefarious drugs [6]. While the pattern of causative factors for opioid use and misuse in the US is well documented [5,7], the identification of key findings from this context can be informative for addressing increased opioid use in the international setting. Studies from the US show that there are two distinct patterns which are interrelated, that of increased prescription of opioids which is associated with a secondary wave of increasing use of illegal opioids such as heroin and fentanyl. Notwithstanding variation within the US, lessons can be learned from the North American context of how a perfect storm of increased prescriptions and the profusion of illegal opioids can lead to a public health disaster [8]. International observations reflecting on increased opioid prescription have widely described the situation as an opioid epidemic, with the effects of over prescribing felt most acutely in the US and Australia [9]. Further complicating the issue of opioid stewardship internationally is the heterogeneity for over prescribing between countries, with reports that sufficiently available opioid analgesia for the treatment of moderate to severe pain refractory to simple analgesia is still a goal and driver for increased opioid prescription in some jurisdictions [10]. From a global perspective, sound opioid stewardship principals and the introduction of international guidelines would contribute to addressing the heterogeneity in the prescription and use of opioids between countries [10]. Subsequently, opioid prescribing practices have drawn public health attention and not solely due to their considerable contribution to community opioid burden. There are also the many opportunities for implementing strategies for the reduction of opioid availability, implementation of opioid stewardship principles and advocacy in a health workforce that is patient centred. Reduction in the number of prescription opioids available in the community has become a public health priority [11].

Reflecting the ubiquitous use and availability of oxycodone IR, this medication is not surprisingly also the most common prescription opioid implicated in opioid poisonings and deaths within Australia, responsible for more deaths and hospitalisations than illicit opioids such as heroin [3,12]. In Australia between 2016 and 17, 3.1 million prescriptions for opioids were dispensed, with a reported 715,000 people using prescription opioids for illicit or non-medical purposes [3]. Notwithstanding the risk that these medications pose, rates of opioid prescribing in Australia have similarly followed the increased prevalence observed in the United States with substantially increased prescription rates since the early 1990s [13]. Worryingly, while rates of opioid prescription peaked in the US in 2011, [5,7] there has been no such peak and subsequent decline identified from available data [13].

These patterns of increasingly high prescription and dispensing rates of opioids are also reported in Australian Emergency Departments (EDs) [14,15]. While the opioid use sequelae of physical dependence and addiction are rare following therapeutic exposure in the emergency department [16], emergency physicians prescribing practices have come under close scrutiny given high rates of opioid prescription intended for outpatient use [17]. Evidence from international studies show that opioid prescriptions from emergency departments make a sizeable contribution to the opioid burden through the availability of unused medications for non-prescription purposes [18]. Diversion describes the acquisition of prescription opioids for use by persons who were not prescribed that medication. This presents a spectrum of dangers, from unsupervised use for otherwise appropriately indicated conditions such as the treatment of moderate pain, through to recreational drug use, addiction and trafficking, which places both the user and the supplier at significant risk. The availability of prescription opiates at risk of diversion is a function of the number of tablets prescribed and dispensed and the number of those tablets consumed by the patient. The proportion of unused oxycodone IR tablets following an ED visit has been reported in one Australian study to remain unchanged when prescription size is reduced [15]. The reason for overprescribing oxycodone IR is multifactorial and may include disproportionate physician concern or a lack of understanding of the implications of overprescribing. The role of emergency care providers in the treatment of acutely painful conditions is juxtaposed with a requirement to ensure opioid stewardship principles are adhered to. Appropriate restriction of access to prescription opioids within the emergency department has been trialled with a variety of interventions reported.

Activities to reduce opioid availability might be considered to fall into one of two categories. First, interventions that aim to monitor prescribing behaviours and to educate or inform prescribers of the harms of opioid oversupply through feedback and reporting mechanisms. This requires individuals to engage with educators or administrators and subsequently make informed choices about the quantities of opioids that they prescribe. This approach of audit and feedback is widely used in healthcare to modify clinicians practice and has been shown to be ineffective in establishing ongoing behavioural change. A Cochrane review and meta-analysis reported that, while some significant effects were observed, audit and feedback resulted in an increase in a desired behaviour of only 4.3% [19]. In the second category are systems and environmental changes, including policy settings, that change the conditions in which the prescribers operate. Adapting interactive systems and physical environments in which healthcare is provided to better adhere to opioid stewardship principals, allows for a permissive and flexible condition in which clinicians can apply their clinical gestalt while the environment facilitates best practice.

Physician education initiatives, including audit and comparison to peers [20,21], opioid prescription monitoring and reporting and opioid prescribing guidelines, Refs. [22,23] have all been employed with varying success rates for reduction of opioids prescribed at local or institutional levels. While policy change, legislation and regulatory bodies have been shown to be effective at scale [24], the latter affords the introduction of elements of implementation science which utilises a broad range of strategies to address both facilitators and barriers of behaviour change. Forcing functions in healthcare are an aspect of Human Factor Engineering, where the design of an interactive system involving people considers the nature of human decision-making behaviours within busy and complex environments and acts to reduce potentially harmful errors. Forcing function works to prevent an action that may unintentionally cause harm but allows that action to be undertaken after the completion of an additional step or action [25].

Health services are therefore increasingly reliant on electronic systems to manage healthcare data and prescribing functions. Software systems provide a platform where an intervention can be automated and therefore uniform across a target group. Prescribing software modifications have been used to study the behaviour of prescribers of opioids in response to both removing pre-populated prescription quantities and incremental changes in the number of tablets prescribed [15,26].

The ubiquitous nature of the opioid epidemic and the heterogeneity of health care provision across the globe raises questions as to the utility of the wider application of interventions, which have shown reductions in opioid dispensing at local or health service levels. Broader approaches and evidence of their effectiveness for reducing opioid prescription medications and dispensing are required.

Statutory regulatory bodies within Australia oversee the monitoring, regulation and funding of medications at a national level and are responsible for approval of therapeutic goods and public funding for subsidised prescriptions under the pharmaceutical benefits scheme (PBS). In June 2020, in response to concerns regarding opioid-associated deaths and hospitalisations, the Pharmaceutical Benefits Advisory Committee (PBAC), advised regulatory changes to the prescription requirements of some opioids, with a reduction in pack sizes of oxycodone IR prescribed for acute pain from 20 to 10 tablets [27]. Following this, changes to default prescription sizes were enacted within the electronic prescribing software for our health service, reducing the prepopulated number of oxycodone IR tablets prescribed to 10, alternate prescription sizes were still available at prescribers’ discretion as were paper-based prescribing methods. This study aimed to examine the effect of the reduced prescription sizes of oxycodone IR auto-populated in electronic prescribing software on the frequency and quantity of tablets prescribed on discharge from the ED, and the associated characteristics of prescriber and patient populations.

## 2. Materials and Methods

This study was conducted in a major tertiary-referral mixed-emergency department in Australia with an annual presentation rate in excess of 75,000 patients. A retrospective review of oxycodone IR prescriptions over two six-month time periods (1 January 2019–30 June 2019 and 1 January 2021 to 30 June 2021) was undertaken, with the two study periods occurring either side of an adjustment to the hospitals prescribing software. The software adjustment reduced the quantity of oxycodone prescribed by auto-populating prescriptions with the number of tablets to be provided, where the default number of oxycodone IR tablets in each prescription was reduced from 20 to 10 tablets between the study periods. Individual physicians retained the ability to manually alter these auto-populated quantities of prescribed oxycodone IR according to their preference. The quantity of oxycodone IR tablets prescribed was compared between the two time periods. The prescribing physicians were identified, and data collected for prescriber’s sex and their years of practice. The years of practice for each prescriber was calculated as the time from their first registration within Australia and the commencement of each study period, these data were obtained from the Australian Health Practitioners Regulation Agency (AHPRA) register. Patient demographic data for age, sex and diagnosis were accessed from the Digital medical record (DMR). Data were collated in Microsoft Excel and checked. Demographic and outcome variables were compared between the two study periods using two-sample t-tests, chi-squared tests and Wilcoxon rank-sum tests where appropriate. We performed a model expansion using a linear mixed effects model where the quantity of oxycodone IR prescribed was the dependent variable. The fixed effects under consideration included the year (2019 or 2021), patient age, patient sex, prescriber sex, hour of treatment and the years of practice of the prescribing medical professional. Random effects included in all models were the type of diagnosis and the individual identifier for prescribers. The models were fit using the lmer package and maximum likelihood in R [28] and model selection was determined by AIC score. The AIC score showed preference for a model with all fixed effects included, aside from hour of treatment and prescriber sex. This preferred model was then refit using restricted maximum likelihood to examine effect sizes and significance of the fixed effect. The study was conducted in accordance with the declaration of Helsinki and approved by the Human Research Ethics Committee of the University of Tasmania (project ID 26167, 22 December 2022).

## 3. Results

A total of 1357 prescriptions for oral oxycodone IR were written over the study period, two records were removed where the prescribed quantity of oxycodone IR was 0. There was an increase in the number of prescriptions observed from 2019 (*n* = 596) to 2021 (*n* = 761). However, when controlled for differential rates of patient presentations between the two study periods, there was no significant difference in the rate of oxycodone IR prescriptions supplied. In total, 596 prescriptions were written in 2019 at a rate of 188.95 per 10,000 presentations, and *n* = 761 prescriptions were written in 2021, a rate of 203.73 per 10,000 presentations. Incident Rate Ratio 0.927 (0.831–1.033) *p* = 0.168. Prescriptions were written by emergency physicians, nurse practitioners and inpatient medical or surgical specialty teams. The number of prescriptions containing 20 oxycodone IR tablets fell from 48.8% (*n* = 291) of prescriptions in 2019 to 8.0% (*n* = 61) in 2021, prescriptions for 10 oxycodone IR tablets increased from 33.7% (201) of prescriptions in 2019 to 71.3% (543) in 2020 (Figure 1).

The median number of oxycodone IR tablets per prescription decreased significantly between the two study periods, from 13.5 to 10.0 tablets (*p* ≤ 0.0001, Table 1). There was an absolute reduction in the number of tablets prescribed between the 2019 and 2021 study periods of 11.67% (Table 1).

Patient demographics were comparable across the study, there was no difference in the age or sex of the patient population or prescriber sex between the two study periods. However, there was a significant difference in the average years of practice of prescribers between the 2019 and 2020 study periods (*p* = 0.046) with a higher average duration of practice observed in 2019 (Table 2).

The maximum likelihood model selection identified a mixed effects model containing year, sex, age and prescriber years of practice to be the best fit for the data. Significant effects were observed for all covariates (Table 3), with an average reduction of 4.55 oxycodone IR tablets per prescription between 2019 and 2021. Patient sex (male), age and increasing years of practice were all observed to increase size of oxycodone IR prescriptions.

## 4. Discussion

Our results showed a significant (*p* ≤ 0.001) reduction in the size of oxycodone IR prescriptions between 2019 and 2021 following changes to default prescription quantities entered in our institutions electronic prescribing software. There was no significant variation measured in the patient populations between the two periods and a minor, although significant, difference in the average years of practice of prescribing medical professionals. Our study confirmed a reduction in prescription size (−4.55, *p* < 0.001). An absolute reduction in the quantity of oxycodone IR tablets prescribed between the two study periods of 11.65% was also achieved. The prescription rate of oxycodone IR per 10,000 patient presentations was not significantly different when controlling for increased patient presentation rates between study periods.

Affecting and sustaining change in healthcare provider practices is difficult, and simply establishing the effectiveness of an intervention is often not enough to sustain the desired change in behaviour [29]. Opioid prescribing in the ED occurs within a complex, busy and dynamic environment that places significant cognitive load on clinicians. While prescribers may espouse sound opioid stewardship principles, better decision making can be assisted through employing non-coercive behaviour modification strategies that still allow clinicians the autonomy to exercise their clinical judgement. Retaining the ability to manually alter prescriptions preserves clinical gestalt when recognizing varied analgesic requirements and pain trajectories, inclusive of patient specific considerations. Reducing the cognitive load of implementing opioid stewardship by utilizing automated software inputs for prescription sizes is a way to ‘nudge’ clinicians towards smaller prescription sizes. The theory of ‘Nudging’ was first adopted from behavioural economics and has been employed to engineer changes in clinician behaviour in many healthcare settings, through novel modifications within electronic medical records and prescription software applications [30]. A reduced number of opioids auto-populated in the prescription software ‘nudges’ the prescriber towards smaller prescription sizes and forces them to manually alter the prescription if intending to increase the number of tablets prescribed. This intermediary step acts as a forcing function, making the prescriber actively reflect on the size of the prescription when deviating from prepopulated values.

Clinicians are more likely to prescribe the default dosage pre-filled in prescribing software [31,32], finding more effort is required to alter a prescription and a belief that the default is an implicit recommendation that reflects best practice [33]. This phenomenon can be used in health policy to influence healthcare outcomes on a broad scale. Previous studies examining the effect of removing prepopulated prescription quantities from prescribing software reported a reduction in prescription size of comparable magnitude to our findings [26]. Our study differs in that a greater reduction in absolute prescription size was achieved by specifying the smaller number of tablets. An absolute reduction in excess opioids has increased benefit, as the proportion of unused opioid analgesia retained by patients post discharge has been shown to remain unchanged with prescription size [15]. Differentiation between prescriptions intending to provide a specified duration of analgesic effect (e.g., three days) and those of a preset quantity of tablets has shown that a large variation in opioid consumption is seen across a range of painful conditions [18]. Canadian [18] and US [34] studies examining opioid pill consumption on discharge from EDs calculated that analgesic requirements to cover the needs of 95% of the population for undifferentiated acute pain for three days was 75 MME. A prescription of 10 oxycodone IR 5 mg tablets equates to 75 MME. Following prescribing software changes in this study, an average of 73.95 MME per prescription was achieved and suggests the change was an appropriate adjustment to prescription size. While further research needs to identify factors contributing to unused opioid prescriptions, smaller prescriptions will potentially reduce the absolute number of tablets available for diversion or non-prescription use in the community.

Our multivariate analysis found patient age, patient sex and prescriber years of practice had a small yet significantly additive effect on the size of discharge oxycodone IR prescriptions. While patient age and sex are frequently included in analyses of prescribing practices, their reported effects vary. The increase in oxycodone prescription size associated with increasing patient age found in our study contrasts with reports of larger discharge prescriptions provided to younger patient populations [35,36]. Age-related disparities in opioid analgesia are likely to be multifactorial and demographic specific. Previous international studies report increased discharge opioid quantities in younger cohorts presenting with musculoskeletal pain [35] and analysis of a variety of specific individual presenting complaints reporting a higher level of pain and analgesia associated with younger age and female sex [36]. A reversal of this trend has been reported in Australian studies, and is seen in our results, with larger quantities of opioids prescribed in older cohorts [3]. Opioids present different risks for older people, with complications of polypharmacy and drug–disease interactions magnified in this group [36,37]. Conversely, an unwillingness to provide discharge prescriptions for opioid analgesia to older patients, perceived to be at an increased risk from these medications or due to a desire to reduce opioid prescribing globally, may lead to an underutilization of appropriate analgesics and leave patients in significant discomfort or distress [38]. Optimal management of pain relief for older patients on discharge from an acute hospital presentation presents a complex challenge for medical practitioners with further research needed to inform prescribing best practice.

This study found that the male sex was associated with increased prescription size. However, this contrasts with several studies that report younger, female patients disproportionately receiving more opioids during emergency presentations [36,39,40]. Prescribing variability, attributed to the prescriber’s sex and its interaction with the patient’s sex has been previously documented [41]. In our analysis, we incorporated prescriber sex as a covariate; however, the model selection process demonstrated that its inclusion resulted in less optimal fit when compared to the preferred model. Further investigation is required to explain increased opioid prescribing to males in this cohort. We found a small yet significant increase in prescription size associated with increased prescriber years of practice. The effect of prescriber experience has been reported with varied results both in the length of practice and with familiarity with ED prescribing practices. Reduced rates of opioid prescribing by ED-attending physicians and residents have been reported when compared with non-ED residents [42], suggesting that specialty training and familiarity with the ED environment have an impact on prescriber behaviour. Our study confirms findings that more experienced clinicians prescribe increased amounts of opioids on discharge [35].

Some limitations were evident in this study. The reduced quantity of oxycodone IR auto-populated into prescriptions was the result of alterations of prescribing software in response to PBS billing arrangements nationally. This did not allow the researchers to undertake a comparative interrupted time series analysis which would have added increased methodological rigor to the study. Nor did it allow the experimental alteration of prescription sizes to explore optimal prepopulated values. This study examined prescription of oxycodone IR in a single ED and did not compare prescription of other opioids at this site or with other emergency departments. This study also measures the quantity of oxycodone IR prescribed. It is therefore possible that variations in the quantity of oxycodone dispensed may have occurred due to differential pharmacist practice when dispensing the medications, although the primary outcome was measuring reduction in prescription size. The calculation of years of practice, which was a function of the date of first registration as a medical health practitioner within Australia and date of commencement of each study period, may not account for the date of commencement of a specialty training program, or accurately reflect the experience of overseas trained physicians who recently registered to practice within Australia. Further variation might be attributable to years of practise due to varying educational priorities for physicians over time, this requires further investigation. This study was also limited to electronic prescriptions and did not capture handwritten paper prescriptions.

## 5. Conclusions

We observed a 11.67% absolute reduction in the number of oxycodone IR tablets prescribed and an average reduction of 4.55 tablets per prescription between the two study periods. Using electronic prescribing software to automate smaller prescription sizes is effective in achieving significant reductions in the number of oxycodone IR tablets prescribed on discharge from the emergency department. Further research is required to confirm the effects of age, sex and prescriber experience on oxycodone IR prescribing practices.

## Figures and Tables

**Figure 1 pharmacy-12-00044-f001:**
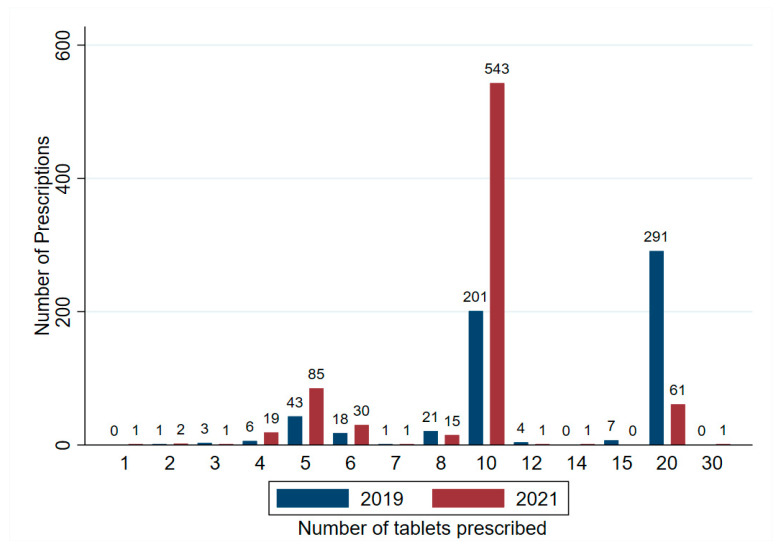
Frequency of oxycodone IR prescriptions and quantity for tablets per prescription following changes in electronic prescribing software in an Australian tertiary ED, prefilled values of oxycodone IR were reduced from 20 (2019) to 10 (2019).

**Table 1 pharmacy-12-00044-t001:** Number of oxycodone IR prescriptions and tablets prescribed on ED discharge pre (2019) and post (2021) reduction in electronic prescribing software default prescription size from 20 to 10.

Year	No. Oxycodone IR Prescriptions	No. Tablets Prescribed	Morphine Milligramequivalent (MME) ^1^	Mean ^1^	Median ^1^
2019	596	8516	107.17	14.20	13.5
2021	761	7522	73.95	9.86	10
Two sample Wilcoxon rank-sum			*p* ≤ 0.0001	Z = 13.502	

^1^ per prescription.

**Table 2 pharmacy-12-00044-t002:** Population demographics by year.

	2019	2021	Total	
Prescriptions	596	761	1357	
Demographics				
Age				
Mean, (years ± SD)	45.20 (18.19)	45.77 (17.76)	45.52 (17.95)	*p* = 0.559 ^1^
No. Patients ≤ 14, *n* (%)	2 (0.16)	4 (0.53)	6 (0.44)	-
No. Patients ≥ 65, *n* (%)	103 (17.28)	135 (17.74)	238 (17.54)	-
Patient Sex, *n* (%)				
Male	301 (50.50)	352 (46.25)	653 (48.12)	-
Female	295 (49.50)	409 (53.75)	704 (51.88)	*p* = 0.12 ^2^
Prescriber Sex, *n* (%)				
Male	297 (49.83)	348 (45.73)	645 (47.53)	-
Female	299 (50.17)	413 (54.27)	712 (52.47)	*p* = 0.133 ^2^
Years of Practice				
Mean, (years ± SD)	6.17 (8.52)	5.39 (5.43)	5.74 (6.97)	*p* = 0.046 ^1^
Years (min–max)	0.20–37.35	0.48–33.50	0.20–37.35	-
0–1 years, *n* (%)	141 (23.98)	133 (17.92)	274 (20.60)	-
>1–10 years *n* (%)	349 (59.35)	485 (65.36)	834 (62.70)	-
>10–20 years *n* (%)	68 (11.56)	109 (14.69)	177 (13.30)	-
>20–30+ years *n* (%)	30 (5.11)	15 (2.03)	45 (3.40)	-

^1^ Two sample *t*-test ^2^ Chi-squared test.

**Table 3 pharmacy-12-00044-t003:** Linear mixed effects model, prescriber ID and diagnosis were included in model as random effects.

Variable	Coefficient	Std. Error	T–Statistic	Prob.
Intercept	12.65	0.47	26.93	<0.001
Year (2021)	−4.55	0.34	−13.227	<0.001
Patient sex (male)	0.50	0.23	2.194	0.028
Patient age	0.02	0.0064	3.076	0.0021
Years of Practice	0.14	0.034	4.127	<0.001
Number of Observations	1332			
AIC	7649.051			

## Data Availability

The data presented in this study are available on request from the corresponding author. The data are not publicly available due to privacy laws in the country of origin.

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
