# Peer review of "Influences of Software Changes on Oxycodone Prescribing at an Australian Tertiary Emergency Department: A Retrospective Review"

_pharmacy, 2024, doi:10.3390/pharmacy12020044_

Round 1
Reviewer 1 Report
Comments and Suggestions for Authors
The manuscript “Influences of software changes on Oxycodone prescribing at an Australian Tertiary Emergency Department” is well-written and carefully analyzed. The intro thoroughly sets-up the key issues.
General concerns
A. Figure 1 is very non-informative. The key information is there but the audience will struggle to make any inferences. Instead of a raw frequency, could the data be expressed as the percentage of the total within that year? Consider trying some alternative plots (e.g. histograms with data in larger bins (0 to 9, 10 to 19, 20+ so as not to be such an overwhelming amount of information?).
B. Pre-post designs with no comparison group are inferentially weak. Do you have ready access to a comparison opioid (e.g. oxycodone slow release? Or morphine?)? If the prescribing for that did not change during the same period, that would add a lot! If this information is not readily unavailable, the discussion should include the national pattern in prescription opioids in Australia from 2019 to 2021. The limitations paragraph (L 321) should also include to a nod that there was no comparison drug (or comparison institution) and there could be alternative factors that account for the observed reductions.
Minor points
Line 4: This is just a light (friendly) suggestion. Pubmed indexes by middle names. Additional authors should consider whether they’d like to list a middle-initial.
L 13: Consider reserving the term “significant” in scientific writing as short-hand for “statistically significant”. important?
L 18: Some style guides indicate that this should be written in the past tense (investigated)
L 26: Please report data consistently to the tenths (10.X%).
L 33: Light suggestion: An informationist indicated that the keywords are for Pubmed and other database indexing. These will already pick up the title terms so there’s no need to repeat oxycodone or emergency department
L 37: opioids (lower-case)
L 41: Of these<comma>
L 49: Light suggestion. The term “trend” has different meanings for the lay public vs statisticians. Journalists use it to mean pattern in the data. Some statisticians use it to mean a non-significant finding (p > .05 but < .10, ick!). Consider “escalation” instead.
L 50: “increase in prescription opioids for the past two decades”? Prescription opioids in the US peaked around 2011 and have been going down since then. This statement needs to be clarified.
L 54: Again, consider “While the pattern…. of causative factors” instead of “trend”.
L 55: well-documented (consider adding a few citations, perhaps: DOI: 10.1097/j.pain.0000000000002473 & doi: 10.1016/j.amepre.2018.01.034 & doi: 10.1016/j.amepre.2021.10.022.
L 57: consider use of thesaurus for “trends” L 61 & 81 too.
L 66: perspective<comma>
L 70: consider replacing “significant” with another term (e.g. appreciable? substantial?). L 94 too.
L 82: Again, US prescribing rates increased from 90s to 2011 and then have declined. This needs to be clarified.
L 87: citation(s)? If available?
L 119: The language so far has been very physician-centric. Do NPs (line 197) in Australia prescribe an appreciable subset of opioids? If so, consider giving an occasional nod to them in the intro too.
L 133: therefore
L 143: medications<delete .>
L 188: 2022).
L 194: periods<comma>
L 199: Can you double-check these #s? 291 = 48.8% in 2019 but 61 = only 0.13% in 2020?
L 207: Can this be reported to the tenths, even if 10.0%? The < symbol should be underlined.
L 210 & 219: The tables should be stand-alone so that the interested reader that is just reading those would have a reasonable understanding of what was done. Consider adding more information (e.g. Country of data collection, why 19 & 21 are important)
L 220: the second super-script 2 would be less-confusing if it were spelled out “square” instead
L 221: Are you studying sex or gender? https://www.coe.int/en/web/gender-matters/sex-and-gender
For Table 2, its possible that different software could produce slightly different results. Still, GraphPad Prism produced a p = .1331 instead of .134 for sex of prescriber.
L 231: Can the number of observations and AIC information go somewhere else? It looks odd dangling there.
L 233: showed
L 246: replace significant with sizable or another term
L 278: study<comma>
L 316: Yes, years of practice corresponds with experience. Years of practice also corresponds with when one received their formative education. Is it also possible that younger physicians education was more concerned with excesses in opioid prescribing? Consider exploring this a bit more.
L 323: nationally<period> This
L 332: run-on sentence
L 339: Significant is a pretty low bar. Consider also listing the % reduction.
Is oxycodone with naloxone (or naltrexone) approved for use in Australia? If so, consider whether that and ER could be an area for future opioid stewardship research.
Refs: If the journal names are abbreviated per journal style, this should be done consistently.
Reviewer 2 Report
Comments and Suggestions for Authors
1. Add study design to the title
2. Shorten the abstract and make it more concise
3. Line 54 - introduce abbreviation US on the first mentioning
4. Introduction is too long and it is missing a flow. Please try to improve it
5. Line 82 - what is F at the end of a sentence
6. Add aim at the end of the introduction
7. Line 196 - sentence does not have a verb
8. Figure 1 - what is frequency? Improve y axis
9. Explain in limitation why year 2023 is not included
Comments on the Quality of English LanguageTypos and missing words
Round 2
Reviewer 1 Report
Comments and Suggestions for Authors
The majority of the prior concerns have been sufficiently addressed to warrant publication.
Reviewer 2 Report
Comments and Suggestions for Authors
I believe the manuscript is now acceptable for publication